# Sirtuins: Promising Therapeutic Targets to Treat Ischemic Stroke

**DOI:** 10.3390/biom13081210

**Published:** 2023-08-01

**Authors:** Yue Liu, Liuding Wang, Guang Yang, Xiansu Chi, Xiao Liang, Yunling Zhang

**Affiliations:** 1Xiyuan Hospital, China Academy of Chinese Medical Sciences, Beijing 100091, China; bettermely@163.com (Y.L.); liudingwang97@163.com (L.W.); chixiansulucky@163.com (X.C.); 2Guang’anmen Hospital, China Academy of Chinese Medical Sciences, Beijing 100053, China; gabrielyg@163.com

**Keywords:** sirtuins, ischemic stroke, deacetylation, neuroprotection

## Abstract

Stroke is a major cause of mortality and disability globally, with ischemic stroke (IS) accounting for over 80% of all stroke cases. The pathological process of IS involves numerous signal molecules, among which are the highly conserved nicotinamide adenine dinucleotide (NAD^+^)-dependent enzymes known as sirtuins (SIRTs). SIRTs modulate various biological processes, including cell differentiation, energy metabolism, DNA repair, inflammation, and oxidative stress. Importantly, several studies have reported a correlation between SIRTs and IS. This review introduces the general aspects of SIRTs, including their distribution, subcellular location, enzyme activity, and substrate. We also discuss their regulatory roles and potential mechanisms in IS. Finally, we describe the current therapeutic methods based on SIRTs, such as pharmacotherapy, non-pharmacological therapeutic/rehabilitative interventions, epigenetic regulators, potential molecules, and stem cell-derived exosome therapy. The data collected in this study will potentially contribute to both clinical and fundamental research on SIRTs, geared towards developing effective therapeutic candidates for future treatment of IS.

## 1. Introduction

Stroke is a major cause of global mortality and morbidity, with two-thirds of stroke survivors developing disabilities, which pose a heavy burden on families and society [1]. Ischemic stroke (IS), which accounts for more than 80% of all strokes, is the most common type of stroke [2]. The sudden cessation of cerebral blood flow results in a complex cascade of events that lead to neuronal cell death and brain damage. The current and the only approved medical therapy for ischemic stroke is the recombinant tissue plasminogen activator (rt-PA); however, it should be administered within 4.5 h of stroke onset [1,3]. Mechanical thrombectomy, an alternative or additional method to rt-PA, has expanded the treatment time window producing therapeutic effects [4,5]. Nonetheless, ischemia-reperfusion injury after these clot-targeting therapies may lead to serious neurological deficits and prognosis. Therefore, it is crucial to elucidate the underlying mechanisms of brain damage associated with ischemic stroke and develop novel therapeutic targets.

Recent evidence suggests that histone modifications regulate the pathological mechanisms that cause ischemic stroke [6,7]. Histone acetylation, dynamically regulated by histone acetyltransferase (HATs) and histone deacetylase (HDACs), is important for global changes in gene transcription and protein expression [8,9]. Sirtuins (SIRTs) are a class of nicotinamide adenine dinucleotide (NAD^+^) dependent HDACs with a highly conserved central catalytic core. Increasing NAD^+^ levels can improve the activity of SIRTs. The sirtuin family has seven members, i.e., SIRT1-SIRT7. They have highly conserved NAD^+^ binding and catalytic domains [10]. The various subcellular localizations, binding sites, and functions of SIRTs are partially explained by differences in their sequence and length in their N and C-terminal domains [11]. SIRTs modulate cell differentiation, energy metabolism, DNA repair, inflammation, and oxidative stress by catalyzing NAD^+^-dependent deacetylation, desuccinylation, and ADP glycosylation reactions [10,12,13]. SIRTs have been associated with neurodegenerative diseases, cardiovascular diseases, diabetic kidney disease, and other aging-related diseases [14,15,16]. Importantly, recent preclinical and clinical studies have reported a correlation between SIRTs and IS, suggesting that SIRTs may be a promising target for IS treatment [17,18,19].

In this review, we present a systematic overview of recent advances in SIRTs research in ischemic stroke and discuss their therapeutic potential. We hope to generate novel ideas for the development of clinical therapy and medications for ischemic stroke treatment.

## 2. Sirtuins in Ischemic Stroke

### 2.1. General Aspects Regarding Sirtuins

#### 2.1.1. SIRT1

Among the sirtuin family members, SIRT1 has been extensively studied and is expressed in a wide range of human tissues, including the brain, liver, muscle, endothelium, pancreas, and adipose tissue [20,21,22]. In the central nervous system, SIRT1 is widely expressed in neurons of the cortex, hippocampus, cerebellum, and thalamus [23,24]. SIRT1 expression at the subcellular level, i.e., in mitochondria, centrosomes, Golgi bodies, and nuclei, varies depending on the tissue and cell type, and occurs through nucleocytoplasmic shuttling, along with growth, development, and stress responses [25]. SIRT1 is expressed in the nucleus and cytoplasm of cultured brain vascular endothelial cells, mainly in the nucleus [26]. The biological effects of SIRT1 are largely attributed to its capacity to deacetylate target proteins. SIRT1 has been shown to deacetylate histone H4 lysine 16 (H4K16) and H3K9, leading to the subsequent suppression of transcription [27]. Moreover, SIRT1 interacts with and deacetylates H1K26, H3K14, H3K18, and H4K6 [27,28]. Other than the histone deacetylation, SIRT1 can drive the deacetylation of various non-histone proteins, including p53, forkhead-box transcription factor (FOXO), proliferator-activated receptor γ coactivator 1α (PGC-1α), nuclear factor kappa B (NF-κB), hypoxia-inducible factor-1a (HIF-1α), nuclear factor E2-related factor 2 (Nrf2), and AMP-activated protein kinase (AMPK) [29,30]. The broad range of targets influenced by SIRT1 endows it with a critical role in regulating various biological processes, including cellular differentiation, inflammation, oxidative stress, cell apoptosis, mitochondrial function, and autophagy [31,32,33].

SIRT1 exerts a neuroprotective effect in IS. In an experimental study, neurons in the ipsilesional cortex of a mouse brain expressed more SIRT1 upon exposure to middle cerebral artery occlusion (MCAO). Furthermore, treatment with a SIRT1 activator could further reduce the infarct volume [34]. Similarly, clinical studies have shown that serum SIRT1 concentrations in patients with acute ischemic stroke (AIS) are significantly higher than normal and SIRT1 may act as a possible diagnostic marker for AIS patients [35]. The increased expression of SIRT1 after cerebral ischemia may represent an endogenous defense mechanism as a stress response. Additionally, evidence suggests that mice overexpressing SIRT1 show minimal hippocampal damage and preserved cerebral blood flow after bilateral common carotid artery stenosis (BCAO) [36]. However, SIRT1 knockout (KO) mice showed larger infarct volumes after ischemia [34].

#### 2.1.2. SIRT2

SIRT2 is the first sirtuin to be identified; it is extensively expressed in multiple tissues, especially in metabolically active ones, like the brain, liver, heart, and adipose tissue [37,38]. Although SIRT2 mainly localizes in the cytoplasm, it can shuttle to the nucleus under certain conditions to participate in a series of biological processes [38,39]. Existing findings have shown that SIRT2 is specifically enriched in brain oligodendroglia [40]. In MCAO mice, SIRT2 mRNA induction was discovered in the entire hemisphere following the stroke and highly expressed in myelin formed by oligodendrocytes [41]. Similar to this, upregulated SIRT2 expression and nuclear translocation in cerebral stroke patients’ ischemic penumbra were identified [42].

Unlike SIRT1, SIRT2 appears to have a negative impact on IS. SIRT2 inhibition increases neural tolerance to oxidative stress by enhancing the activation of cytoprotective genes [43]. SIRT2 was upregulated during neuronal ischemia in the oxygen-glucose deprivation (OGD)-induced cell model and the MCAO-induced mouse model; notably, downregulating SIRT2 could significantly protect neurons against cerebral ischemia [42]. According to Lea’s study, SIRT2-deficient mice exhibited significantly reduced neurological impairments upon MCAO compared to wild-type animals [41]. Moreover, the level of serum SIRT2 significantly increased in AIS patients compared to controls and negatively correlated with patient prognosis [44].

#### 2.1.3. SIRT3

SIRT3 is predominantly found in the mitochondrial matrix and is widely expressed in highly metabolic tissues such as the brain, heart, liver, and brown adipose tissue [45,46]. Studies indicate that SIRT3 is also present in the nucleus and cytoplasm [47,48,49]. Mitochondria, i.e., the energy center of cells, manufacture adenosine triphosphate (ATP) to fuel cells and are also essential for regulating the production of reactive oxygen species (ROS), maintaining Ca^2+^ homeostasis, controlling osmotic pressure, and transmitting cell signals. SIRT3 is essential for mitochondrial energy metabolism, substrate oxidation, and mitophagy through its deacetylation activity. By deacetylating ATP synthase proteins via SIRT3, acetylome signaling helps maintain energy balance in the mitochondria [50]. Moreover, SIRT3 increases the activity of the antioxidant enzyme superoxide dismutase 2 (SOD2) by deacetylating FOXO3a, resulting in ROS reduction and protecting cells from oxidative stress [51]. However, SIRT3 deficiency can lead to disruption of mitochondrial homeostasis and inhibit mitochondrial autophagy [52].

SIRT3 plays an important role in the regulation of ischemic stroke processes. Previous studies have shown that upregulated SIRT3 expression reduces the infarct volume and improves neurological function, via the SIRT3-FOXO3a-SOD2 pathway in MCAO mice [17]. FOXO3a, which inhibits astrocyte proliferation and cytokine-mediated astrocyte proliferation, is also a crucial mediator of astrogliosis [53]. Astrocyte activation can form a scar after a stroke. Reports indicate that SIRT3 deficiency attenuates the inhibitory effect of adjudin’s (a SIRT3 activator) in the formation of the glial scar, whereas SIRT3 overexpression could further reduce the glial scar by inhibiting astrogliosis through the SIRT3- FOXO3a pathway activation [54].

#### 2.1.4. SIRT4

SIRT4 exists in the mitochondrial matrix of several organs, including the brain, heart, kidney, liver, and muscle [55,56,57]. Although SIRT4 has little deacetylation activity, it functions primarily as an ADP ribosyltransferase [58]. SIRT4 transfers an ADP-ribosyl group from NAD^+^ to the glutamate dehydrogenase enzyme (GDH), which in turn represses GDH activity [55]. Another significant role of SIRT4 is its regulation of glutamine metabolism and promotion of the damage response of deoxyribonucleic acid (DNA) [59]. Furthermore, SIRT4 can regulate the expression of genes involved in fatty acid oxidation in muscle mitochondria and hepatocytes [60].

Although information regarding the role of SIRT4 in ischemic stroke is limited, its function can still provide valuable insights. Excessive glutamate accumulation in neurons causes excitotoxicity, which is the underlying cause of brain injury following IS. Studies have shown that SIRT4 deficiency causes a more severe response to the potent excitotoxin kainic acid, whereas SIRT4 overexpression can increase the protein levels of glutamate transporter 1 (GLT-1) and inhibits the level of glutamine synthetase, thereby preventing excitotoxicity [61,62]. Furthermore, SIRT4 was downregulated in healthy human umbilical vein endothelial cells (HUVECs) during acute hypoxia, resulting in increased oxidative stress and inflammation [63]. These results imply that SIRT4 may be a potential new target for the therapy of acute vascular events including IS.

#### 2.1.5. SIRT5

SIRT5 is expressed widely in various organs, including the brain, heart, liver, kidney, muscle, and testicles, primarily located in mitochondria, but also present in the cytoplasm and nucleus [64,65,66]. In the human brain, SIRT5 transcripts are expressed in all cortical layers, with layer II exhibiting the highest levels [67]. Unlike other sirtuins, SIRT5 has weak deacetylase activity but possesses strong lysine desuccinylation, desmalonylation, and desglutarylase activity due to the larger lysine acyl binding pocket in its structure [68,69]. SIRT5 can reduce excessive ROS levels by binding and succinate SOD, thereby enhancing cells’ ability to resist oxidative stress [70].

The role of SIRT5 in ischemic stroke remains controversial. Studies have shown that SIRT5 maintained mitochondrial respiration and protected against metabolic stressors and cell death after cerebral ischemia [71]. However, some researchers reported that compared to wild-type mice, infarct volume decreased and neurological impairments improved 48 h after 45 min of MCAO in SIRT5 KO mice [72]. Furthermore, upregulated levels of SIRT5 gene expression were observed in peripheral blood monocytes (PBMCs) of AIS patients 6 h after the initial stroke compared to healthy controls [72]. Additional research is necessary to elucidate the precise role of SIRT5 in the brain after ischemia.

#### 2.1.6. SIRT6

SIRT6 is primarily located in the nucleus of the liver, brain, thymus, muscle, and heart and deacetylates histone H3 at lysine sites 9, 18, and 56 sites in a site-specific manner [73,74]. However, the first enzymatic activity discovered in SIRT6 is its NAD^+^-dependent mono-ADP-ribosyltransferase activity [75]. Additionally, SIRT6 can deacetylate various non-histone substrates, such as FOXO1, histone acetyltransferase 5 (GCN5), and C-terminal binding protein interacting protein (CTIP) [76,77,78]. By regulating the activity of key proteins, SIRT6 plays an important role in maintaining genome stability and telomere integrity, promoting DNA repair, preventing aging, and maintaining glucose homeostasis [79].

SIRT6 is highly expressed in the cortex and hippocampus of the brain [80,81]. It protects the brain from DNA damage, neurological impairment, and neurodegeneration [82]. SIRT6 KO brains showed accumulated DNA damage and increased apoptosis [82]. Luca and colleagues constructed the MCAO injury model in vascular endothelial cell (VEC)-specific SIRT6 knockout mice and discovered that these mice exhibited increased cerebral infarction volume, increased neuronal mortality, and aggravated neurological function damage [18]. Conversely, post-ischemic SIRT6 overexpression mitigated neurological deficiency and infarct size [18]. Several studies suggest that SIRT6 is a protective molecule in IS; however, other studies maintain opposing opinions. For example, Shao et al. reported that suppressing SIRT6 reduced oxidative stress-induced neuronal damage [83].

#### 2.1.7. SIRT7

SIRT7 is primarily located in the nucleolus, although evidence suggests that it also exists in the cytoplasm of primary fibroblasts [84]. All mouse tissues investigated expressed SIRT7 mRNA, except for skeletal muscle. In non-proliferating tissues, such as muscle, heart, and brain, SIRT7 levels were very low, but they were high in metabolically active tissues such as the liver, spleen, and testicles [85]. SIRT7 is involved in various cellular processes, including mitochondrial homeostasis, genomic stability, chromatin regulation, and ribosome biogenesis [86].

Studies on the function of SIRT7 in ischemic stroke are scarce. Using high-throughput databases and a stroke systems biological model, Wong et al. identified SIRT7 as a potentially effective target for first aid and emergency treatment within 24 h post-stroke [87]. Lv et al. reported that after OGD reoxygenation treatment, SIRT7 expression in neurons significantly increased, whereas SIRT7 knockout aggravated the OGD-induced injury [88] (Figure 1).

### 2.2. Regulation of Sirtuins in Ischemic Stroke

#### 2.2.1. Energy Metabolism

The brain primarily relies on glucose to produce energy [89], and impaired energy metabolism is a key pathological sign of IS [90,91]. An abrupt reduction in cerebral blood flow causes decreased glucose and oxygen delivery, insufficient NAD^+^ synthesis, and a drop in the NAD^+^/NADH ratio, all of which pose substantial risks to the survival and function of neurons. Mitochondria are the major sites for ATP production, and the integrity of their structure and function is the basis for smooth energy metabolism [92].

ATP synthesis in the electron transport chain is mildly uncoupled by uncoupling protein 2 (UCP2), and high levels of UCP2 expression may prevent cells from producing ATP [93]. Preconditioning with a SIRT1 agonist can significantly reduce the area of cerebral infarction in mice with focal cerebral ischemia, and its mechanism is related to the downregulation of mitochondrial UCP2 [94]. Kevin and colleagues discovered that in neurons of SIRT1 knockout mice, the rate of glycolysis decreased and ATP production was impaired [95]. AMPK is an essential factor that regulates mitochondrial biogenesis and plays a crucial role in regulating energy metabolism and mitochondrial function via proliferator-activated receptor gamma coactivator-1α (PGC-1α) signaling [96,97]. AMPK and SIRT1 can activate each other and have a synergistic relationship in response to ischemic injury [98]. Wan and colleagues reported that AMPK activator pretreatment reduces ATP energy consumption during cerebral ischemia by activating SIRT1, thus showing beneficial effects in protecting neurons [99]. Other studies have shown that SIRT3 overexpression can promote mitochondrial biogenesis and induce autophagy by activating the AMPK pathway, thereby alleviating OGD-induced neuronal damage [100].

Protein kinase C epsilon (PKCε) is an important signaling molecule that protects mitochondrial function and nerve cells and plays a vital role in reducing ischemic injury [101,102]. PKCε enhances the expression of nicotinamide phosphoribosyl transferase (Nampt), an enzyme primarily expressed in neurons that catalyzes the rate-limiting step of NAD^+^. The lack of Nampt after cerebral ischemia can exacerbate the aging of nerve cells [103]. Morris-Blanco et al. discovered that Nampt was integral to PKCε-mediated maintenance of mitochondrial respiration. The activation of PKCε can enhance the desuccinylation of SIRT5. After the deletion of SIRT5, the oxygen consumption in mitochondria significantly reduced, and PKCε could not prevent cortical degeneration after cerebral ischemia injury, strongly suggesting that SIRT5 plays an important role in the regulation of mitochondrial energy generation and the neuroprotective effect after cerebral ischemia mediated by PKCε [71]. Collectively, these studies indicate that SIRTs play an important role in maintaining energy metabolism.

#### 2.2.2. Oxidative Stress

The brain tissue contains high levels of lipids, including unsaturated fatty acids, and possesses weak antioxidant defenses, rendering it susceptible to oxidative stress caused by oxygen free radicals [104]. Neurons experience oxygen and energy depletion after ischemic stroke. When free radical generation, including ROS and reactive nitrogen species (RNS), exceeds the endogenous capacity to counteract them, redox equilibrium is disrupted, causing oxidative stress, which causes extensive damage to neurons [105]. Additional effects of oxidative stress include neuroinflammation, apoptosis, disruption of the blood–brain barrier (BBB), and exacerbation of neurological disability [106].

Previous research has demonstrated that SIRT3 overexpression can decrease the infarct volume and improve the neurologic function of MCAO mice. This mechanism could be associated with the deacetylating of FOXO3a, increasing the activity of the antioxidant enzyme SOD2 and reducing ROS production [17]. Similarly, recent evidence indicates that SIRT3 overexpression effectively attenuates ROS overproduction induced by oxygen–glucose deprivation/reperfusion (OGD/R) in HUVECs [107].

Nuclear factor erythroid 2-related factor 2 (Nrf2) is a redox-sensitive transcription factor that is broadly expressed and regulates the expression of a range of antioxidant genes by interacting with antioxidant response elements (AREs) in the nucleus [108,109,110]. The Nrf2 signaling system is an important antioxidant defense system that removes overproduced free radicals. Increasing evidence suggests that Nrf2 expression upregulates after an ischemic stroke to protect against damage caused by excessive ROS generation [111,112]. However, the absence of Nrf2 exacerbates neurological impairments and cerebral infarction [113,114]. Nrf2 protein and its downstream antioxidant proteins, including HO-1 and NQO1, were upregulated in both the OGD/R and MCAO models [115]. However, SIRT1 silencing blocked these effects and exacerbated the oxidative stress that follows an ischemic stroke. SIRT1 may exert beneficial effects after cerebral I/R by upregulating the Nrf2 protein [115]. SIRT6 was also found to have an antioxidant effect by promoting Nrf2 expression. SIRT6 expression in the cerebral cortex decreased after cerebral I/R, whereas SIRT6 overexpression using in vivo gene transfer improved Nrf2 signaling, decreased oxidative stress, and alleviated the neurological impairments and brain tissue damage caused by cerebral I/R [116]. These data suggest that suppressing oxidative stress via SIRTs is a potentially valuable therapeutic strategy for IS in the future.

#### 2.2.3. Autophagy

Autophagy is a vital process by which cells remove damaged proteins and organelles via lysosomes to maintain normal physiological function and homeostasis. The sudden interruption of cerebral blood flow results in energy deprivation, followed by the activation of autophagy with a biphasic effect on ischemic stroke. As a key process of self-repair, moderate activation of autophagy can protect nerve cells and reduce brain damage. However, excessive or insufficient autophagy can cause autophagic cell death and aggravate brain damage [2]. Various autophagy-related (ATG) regulate different stages of autophagy formation, including isolation membrane formation, autophagy maturation, autophagy-lysosome fusion, and autophagy degradation [117,118].

AMPK is a key cellular energy sensor, and its activation can induce autophagy [119,120]. Conversely, the mammalian target of rapamycin (mTOR) is a strong inhibitor of autophagy, and AKT is an upstream signaling molecule that activates mTOR [121]. Evidence suggests that AMPK inhibits mTOR, thereby causing autophagy [122]. SIRT3 protects against neuronal ischemia in OGD-induced cortical neurons by increasing the number of LC3-positive puncta and autophagosomes by interacting with the AMPK-mTOR pathway [100]. Conversely, SIRT6 can activate autophagy by upregulating the number of LC3 and ATG5 by weakening the AKT signal; however, this aggravates neuronal damage induced by oxidative stress [83]. Moreover, SIRT1 deacetylates FOXO1 to inhibit the expression of autophagy-associated factors, improving cerebral injury in MCAO rats [123]. Taken together, the role of autophagy in acute cerebral ischemia remains controversial. Autophagy activation during cerebral ischemia may either have a neuroprotective impact or promote brain damage [119,124]. Additional studies are necessary to elucidate the crosstalk between SIRTs and autophagy.

#### 2.2.4. Neuroinflammation

An inflammatory reaction is a key pathological factor that contributes to neurological impairment in ischemic stroke. The release of a large number of necrotic substances after an ischemic stroke causes peripheral white blood cells to infiltrate into the brain parenchyma. The activation of microglial cells and astrocytes in the brain parenchyma further triggers a series of strong inflammatory cascade reactions, which destroy the BBB, leading to brain edema and neuronal death [125].

Nuclear factor κB (NF-κB), a transcription factor, is crucial in triggering the inflammatory response. TNF-α, IL-1β, and IL-6 are among the inflammatory cytokines and chemokines that are released when NF-κB is activated [126,127]. SIRT1 can inhibit the transcription activity of NF-κB by deacetylation, resulting in a reduction in vascular inflammation [128]. TRAF6 is an essential adaptor protein that connects upstream receptors to downstream effector molecules in cells, allowing for the transmission of multiple receptor signaling pathways on cell membranes and, consequently, exerting biological effects [129]. Following an ischemic stroke, the expression of TRAF6 in brain homogenates increases in a time-dependent manner [130]. The activation of TRAF6 leads to the activation of the NF-κb signaling pathway, which then triggers the inflammatory response. In a mouse model of MCAO-induced stroke, a decrease in the level of SIRT1 protein increased ROS expression, as mentioned earlier [131]. Interestingly, ischemia-induced TRAF6 accumulation and injury were reduced when the ROS levels dropped, whereas a burst of ROS was accompanied by an increase in TRAF6 protein [131]. In short, according to Yan’s report, the decrease in SIRT1 protein levels leads to an increase in ROS production, and the burst of ROS further leads to an increase in TRAF6 protein levels. The disruption of this mechanism may be related to oxidative stress, inflammation, and injury repair processes in MCAO-induced cerebral ischemic injury in mice [131]. The SIRT1-ROS-TRAF6 pathway is a potential therapeutic target against IS.

High-mobility group box 1 (HMGB1) is a DNA-binding protein that exists widely in the nucleus and cytoplasm of eukaryotes, and it can act as an inflammatory trigger when passively released from necrotic cells [132]. As a key histone deacetylase, SIRT6 decreased significantly during cerebral ischemia, promoting the release of HMGB1 [133]. This was also confirmed in SH-SY5Y neuronal cells induced by OGD [133].

Microglia play a dual role in cerebral ischemia. On the one hand, they produce inflammatory mediators that promote the inflammatory response; on the other hand, they protect neurons from excitotoxicity [134]. Following cerebral ischemia, microglia expressing the fractalkine receptor (CX3CR1) are activated and drawn to the ischemic area. By upregulating CX3CR1, Cao et al. discovered that SIRT3 encouraged microglia to migrate to the ischemic brain [135]. Future research should focus on deeply investigating the complex relationship between SIRT3 and microglia.

Additionally, regulatory T cells (Tregs), a component of the adaptive immune system, serve as an inbuilt defense mechanism to reduce inflammation in the brain following an ischemic stroke. Tregs can prevent the recruitment of peripheral inflammatory cells and the infiltration of microglia in the ischemic penumbra [136]. SIRT2 was significantly upregulated in infiltrating Tregs at day 3 post-MCAO, but this change weakened the anti-inflammatory effect of Tregs [137]. In summary, SIRTs play an important role in regulating neuroinflammation to ameliorate ischemic injury.

#### 2.2.5. Apoptosis

Ischemic neuronal apoptosis is primarily a mitochondrial-centered process that involves genes from the B-cell lymphoma (Bcl-2) family and the cysteine protease family. Anti-apoptotic factor Bcl-2 and pro-apoptotic factor Bax are two crucial regulatory proteins of the apoptotic pathway. During cerebra ischemia, glucose/energy metabolism disorder leads to a decrease in Na^+^/K^+^-ATPase activity, which results in an ion homeostasis imbalance, cell membrane depolarization, and a large influx of Ca^2+^. The high concentration of calcium ions cleaves the Bcl-2 interaction domain (Bid) to produce truncated Bit (tBid), which migrates to the mitochondria and interacts with pro-apoptotic proteins such as Bax, inducing the opening of mitochondrial transition pores (MTP) and the subsequent release of cytochrome c (Cytc) [138]. Cytc induces the transformation of Apaf1 (apoptotic protease activating factor 1), which then binds with procaspase-9 to form an apoptosome, that spontaneously activates the caspase-9 complex. The Cyt-c, Apaf1, and caspase-9 complex continue to activate downstream caspase-3 resulting in the formation of apoptotic bodies. The activated caspase3 hydrolyzes structural and functional proteins related to cell disintegration, leading to DNA fragmentation and apoptosis [139].

Reduction in caspase-3 and other apoptotic markers like Bim and Bad suggest that SIRT2 downregulation may prevent the apoptosis of OGD-induced neurons. The underlying mechanisms may be linked to the suppression of the AKT/FOXO3a and MAPK pathways caused by SIRT2 inhibition [140]. SIRT7 may protect neurons from OGD/R-induced damage by regulating the p53-mediated proapoptotic signal pathway, according to research by Lv et al. [88]. In experimentally generated cerebral ischemic rats, estrogen deficiency caused by ovariectomy exacerbates brain infarction. However, estrogen pretreatment suppresses apoptosis and reduces ischemia-induced cerebral damage. Importantly, the estrogen-induced neuroprotection effect was abolished in SIRT1 knockout mice and AMPK-inhibited rats. It is reasonable to speculate that estrogen protects against ischemic injury through the SIRT1/AMPK pathway, with SIRT1 being a key mediator in this process [141]. Additionally, Yan et al. found that SIRT1 deacetylates NF-κB to reduce the apoptosis rate and improve cerebral infarction conditions in IS rats [142]. It should be noted that, as mentioned earlier, SIRT1 is primarily localized in the nucleus, and its high expression in the nucleus is associated with anti-apoptotic effects, whereas overexpression in the cytoplasm does not possess this anti-apoptotic function [143]. In short, these findings suggest a possible link between apoptosis and SIRTs, involving neuroprotection. This opens up a new avenue for targeting SIRTs in the treatment of cerebral ischemia.

#### 2.2.6. Blood–Brain Barrier Integrity and Angiogenesis

The BBB, an important physical and biochemical barrier between blood and brain tissue, majorly comprises brain microvascular endothelial cells (BMECs) and is wrapped by pericytes and astrocytes [144]. The BBB can protect brain tissue from toxic substances and control the stability of the central nervous system. The destruction of the BBB is one of the significant pathological features of ischemic stroke, playing an important role in promoting neurological deficits [145]. Therefore, it is urgent to develop new therapeutic strategies to preserve the integrity of the BBB after ischemic stroke.

The tight junctions (TJs) proteins expressed by BMECs can restrict cell bypass permeability and are crucial regulators of BBB integrity [146,147]. HIF-1α activation can disrupt TJs and increase BBB permeability [148]. SIRT3 helps in relieving brain edema and BBB damage after ischemic stroke, which might be achieved by increasing TJ proteins ZO-1 and Occludin by regulating the HIF-1α signaling pathway [149]. Diaz-Cañestro’s study revealed that SIRT5 is an important promoter of brain injury and neurological deficits. This effect is mediated by increasing BBB permeability and reducing the expression of TJ proteins through the PI3K/AKT pathway in MCAO mice models and primary human brain microvascular endothelial cells [72]. Additionally, SIRT6 has been found to attenuate BBB damage in an MCAO-induced mice model [18]. Using mice models lacking SIRT6, researchers found that mice lacking SIRT6 exhibited larger stroke size, more severe neurological deficits, and increased blood–brain barrier disruption [18].

There is evidence to suggest that increased microvessel density in the peri-infarct region is associated with longer survival in patients with ischemic stroke [150]. This indicates that enhancement of angiogenesis is one of the effective strategies for promoting functional recovery after ischemic stroke [151,152]. The vascular endothelial growth factor (VEGF) is one of the main targets of HIF-1α and is a potent growth factor that plays an important role in angiogenesis by activating extracellular regulated protein kinases (ERK) [153]. Indeed, the pro-migratory effect of vascular endothelial growth factor (VEGF) has been found to be mediated by the inhibition of thioredoxin-interacting protein (TXNIP) [154]. In the context of cerebral ischemia, energy restriction-induced SIRT6 has been found to inhibit TXNIP transcription and promote angiogenesis, which is critical for tissue repair and recovery after cerebral ischemia [155]. However, VEGF is also a vascular permeability factor that controls paracellular permeability and is responsible for disrupting BBB function [148,156]. Therefore, VEGF plays a dual role following an ischemic stroke [157]. During the acute phase of ischemic stroke, the levels of HIF-1α and VEGF proteins in SIRT3-deficient mice increased, resulting in more severe BBB damage [149]. Moreover, in the I/R model constructed using human BMECs, SIRT3 overexpression reduced the permeability of the BBB by regulating the expression of tight junctions [158]. However, during the recovery period of cerebral ischemia, SIRT3 KO mice showed a notable decrease in VEGF levels and ERK phosphorylation, and the infarct striatum exhibited lower vascular density [159]. Hence, it is important to note that the effectiveness of targeting SIRT3 to enhance angiogenesis as a therapeutic approach for IS recovery may vary depending on the intervention time. Further research is needed to optimize and validate this strategy and determine the long-term benefits and potential risks.

#### 2.2.7. Neurogenesis

In recent decades, several neurorestorative therapies have been developed to activate brain plasticity and support functional recovery in both experimental and clinical investigations. Neurogenesis, which offers essential support for remodeling and restoring brain architecture and function, has attracted significant attention in the last decade [160]. Neurogenesis includes the proliferation, migration, and differentiation of neural stem cells (NSCs). Research has demonstrated that adult neural stem cells located in the subventricular zone and the subgranular zone of the hippocampal dentate gyrus can be activated and generate neuroblasts. These neuroblasts are recruited to the ischemic area and participate in the repair of ischemic brain tissue [161,162,163]. Zhao and colleagues found that SIRT1 and SIRT2 contribute to NSC proliferation, whereas SIRT1, SIRT2, and SIRT6 contribute to NSC differentiation [164]. Specifically, an increase in the proportion of young neurons and a decrease in the proportion of mature neurons, without impacting glial differentiation, have been discovered to be effects of SIRT6 on neuronal maturation and differentiation within the hippocampus [80]. SIRT6 is one of several key players that regulate the complexity of adult hippocampal neurogenesis.

In summary, we discussed the regulatory mechanisms of SIRTs in ischemic stroke, including energy metabolism, oxidative stress, autophagy, neuroinflammation, apoptosis, BBB disruption, angiogenesis, and neurogenesis. We can obtain that SIRTs play important roles in both the acute and long-term stages of post-ischemic stroke. SIRTs can exert a neuroprotective effect in the acute stage following a stroke. They help reduce inflammation, oxidative stress, and neuronal apoptosis, promoting cell survival and preserving the integrity of the BBB. SIRTs also regulate energy metabolism and enhance cellular resilience to ischemic injury. In the chronic stage after a stroke, SIRTs play a regulatory role in neurorepair and angiogenesis, contributing to the promotion of functional recovery. It should be noted that the beneficial effects of SIRTs are not solely attributed to their activation. The role of certain SIRTs, such as SIRT4, SIRT5, and SIRT7, in IS, is still unclear. Targeting SIRTs appropriately holds promise in the treatment of IS (Figure 2).

## 3. Therapeutic Research

### 3.1. Pharmacotherapy

Drug therapy is an important component of modern medicine, which helps individuals combat diseases and improve their health. With the continuous progress of science and technology, researchers are constantly seeking new drugs and treatment methods to address the challenges posed by diseases. In the following section, we will review drug therapies targeting SIRT, including NAD^+^ enhancers, SIRTs activators, and SIRTs inhibitors (Table 1). This section aims to provide valuable insights for further development and innovation in the field of drug therapy.

#### 3.1.1. Mitigating Mitochondrial Injury

Melatonin is a hormone secreted by the pineal gland and a promising neuroprotective agent. Yang and colleagues reported that melatonin treatment activates SIRT1 signaling and thus increases Bcl2 expression, decreases the level of Bax, and attenuates mitochondrial dysfunction in MCAO-induced mice [165]. In addition, melatonin retained the mitochondrial mass affected by OGD/R in mouse hippocampal HT22 cells, and restored the mitochondrial fusion/division kinetics. Interestingly, this phenomenon is accompanied by an increase in SIRT3 levels [166]. Promoting SIRTs activity by increasing the concentration of NAD^+^ is also a beneficial therapy. Research has shown that administering nicotinamide mononucleotide (NMN), a precursor to NAD^+^ synthesis, can significantly improve ischemic brain injury caused by global cerebral ischemia [167]. In a mouse model of cerebral ischemia induced by bilateral common carotid artery occlusion (CCAO), NMN treatment was found to enhance the activity of SIRT3, leading to the suppression of ischemia-induced mitochondrial fragmentation and the generation of ROS. This study has unraveled the importance of NAD^+^ and SIRT3 in maintaining mitochondrial health and functionality [168].

Resveratrol is the first natural compound to be discovered that could activate SIRT1, which plays an important role in protecting against neurological damage. Resveratrol could reduce ATP energy consumption during ischemia to exert a neuroprotection effect by activating the cAMP/AMPK/SIRT1 pathway [99]. Resveratrol’s translatability in the treatment of IS is encouraging in terms of clinical efficacy. A two-period, open-label, single-arm, within-subject control study showed that healthy participants tolerated trans-resveratrol 2000 mg twice daily with an acceptable exposure [169]. Notoginseng leaf triterpenes (PNGL) is a dammarane-type saponin purified from Panax notoginseng stem and leaf. In OGD/R-induced SH-SY5Y cells, PNGL remarkably reversed the decrease in intracellular NAD^+^ and NADH and the depletion of ATP production, reducing neuronal necrosis and improving neuronal survival under ischemia and hypoxia conditions. PNGL may exert neuroprotective effects by regulating the Nampt-NAD^+^- SIRT1 pathway [170].

#### 3.1.2. Anti-Oxidative Stress Damage

Picetannol (Pic) is a natural compound with a strong antioxidant capacity found primarily in seeds, wines, and fruits [171]. Wang et al. reported that Pic treatment exerts strong antioxidant effects by upregulating the activities of antioxidant enzymes SOD, catalase (CAT), and glutathione peroxidase (GSH-Px) by activating the SIRT1/FOXO1 pathway [172]. Diosmetin, a bioflavonoid compound isolated from citrus fruits, displayed antioxidative stress effects on various organs. Administration of diosmetin before MCAO improved neurological outcomes and decreased the cerebral infarct volume and pathological lesions of rats by reducing the levels of lactate dehydrogenase (LDH) and ROS and inhibiting oxidative stress damage via SIRT1/Nrf2 signaling pathway activation [173]. Similarly, mangiferin inhibited LDH release and ROS generation to exert a neuroprotection effect by activating the SIRT1/PGC-1α pathway [174]. Alpha-lipoic acid (ALA) is a natural antioxidant isolated from plants and animals for use as a dietary supplement. ALA has been proven to increase the SOD level to protect the MCAO mice brain against ischemic damage by upregulating SIRT1-dependent PGC-1α [175]. In addition, Zhou and colleagues revealed that kaempferol (KFL), a natural flavonol, significantly upregulated SIRT1 protein and inhibited OGD-induced oxidative stress, resulting in a decrease in the levels of ROS and LDH and an increase in SOD activity and GSH content [176].

#### 3.1.3. Regulating Autophagy

A few natural products exert neuroprotection effects by regulating autophagy. Betulinic acid (BA) is a biologically active pentacyclic triterpene compound extracted from Betula platyphylla. BA pretreatment was reported to notably reduce the ratio of autophagic cells, downregulate autophagy-associated factors Beclin 1 and LC3I/II, and upregulate p62 level in MCAO rats, as well as activate the SIRT1/ FOXO1 signal pathway [123]. In ischemic rats administrated with quaternary aporphine alkaloid magnoflorine, the fluorescence intensity of LC3 was remarkably downregulated, whereas p62, SIRT1, and AMPK were upregulated. Therefore, it is rational to conclude that magnoflorine relieved cerebral ischemia-induced neuronal damage by activating the SIRT1/AMPK pathway [177].

#### 3.1.4. Inhibiting Neuroinflammation

MDL-811 is a superior SIRT6 activator with strong anti-inflammation in microglia/macrophages, which could attenuate brain injury and improve neurobehavioral deficits in MCAO mice by inhibiting the neuroinflammation via the SIRT6/EZH2/FOXC1 pathway [178]. Notably, MDL-811 treatment inhibits the expression of pro-inflammatory factors (TNF-α and IL-1β) in primary human monocytes of patients with ischemic stroke through activating SIRT6 [178]. Moreover, treatment with activator 3 (SIRT1 activator) can significantly reduce cerebral infarction volume in MCAO mice, possibly via inhibition of NF-κB induced inflammation and apoptosis pathways [34].

Research reports have shown that ginkgo biloba (GA) contains flavonoid-like compounds and terpenoids with potent anti-inflammatory, antioxidant, and free radical scavenging properties [179]. Xu et al. revealed that GA treatment ameliorates the neuroinflammation in the brain tissue of stroke rats by targeting the SIRT1/ NF-κB pathway [180]. Studies have shown that arctigenin (ARC), a phenylpropanoid dibenzylbutyrolactone lignan derived from Arctium lappa L, protects against cerebral ischemia injury in rats by inhibiting NLRP3 inflammasome activation and lowering the levels of inflammatory factors IL-1β and IL-18 by activating SIRT1 signaling in the brain [181]. Bergenin (BGN), a C-glycoside of 4-O-methylgallic acid, exerts anti-inflammatory, antioxidant, and tissue repair effects. BGN inhibited the activation of microglia, the phosphorylation of NF-κB, and the expression of inflammatory factors IL-1β, IL-6, and TNF-α. Moreover, BGN upregulates SIRT1 and FOXO3a, suggesting its potential in alleviating IS-mediated neuroinflammation by enhancing SIRT1/FOXO3a pathway [182]. Cycloastragenol can significantly upregulate the expression of SIRT1 in the ischemic brain and inhibit the mRNA expressions of pro-inflammatory factors TNF-α and IL-1β [183]. Besides these compounds, trilobatin (TLB), a small molecule monomer isolated from the Chinese herb *Lithocarpus polystachyus* Rehd, directly interacts with SIRT3 to increase the expression and activity of SIRT3, thereby inhibiting the toll-like receptor 4 (TLR4) signaling pathway and alleviating neuroinflammation after middle cerebral artery occlusion in rats [184]. Inflammatory cytokines and TLR-4 promote NF-κB activation, resulting in severe neurological damage [126].

#### 3.1.5. Anti-Apoptosis

AK-7 is a SIRT2-specific inhibitor, which can remarkably decrease the infarction volume and promote the recovery of neurological function by activating p38 [185]. Additionally, SIRT2 inhibitors AK-1 and AGK2 have been proven to reduce apoptotic cell death by SIRT2 inhibition in OGD-induced cortical neurons [140].

Resuvastatin is a synthetic statin, which lowers lowering lipid levels, and exerts anti-inflammation as well as anti-oxidation. Resuvastatin may reduce cerebral infarction area and apoptosis rate in IS rats; the underlying mechanism may be related to SIRT1/NF-κB pathway regulation [142]. Melatonin has also been proven to increase SIRT3 expression and, via activating SIRT3, reduce neurological impairment and cell apoptosis after transient middle cerebral artery occlusion (tMCAO) in mice [186].

In MCAO rats, resveratrol pretreatment could enhance the SIRT1 activity, thus decreasing the levels of p53 and caspase3, and improving ischemic stroke by inhibiting cell apoptosis [187]. Calycosin-7-O-β-D-glucoside (CG), a representative isoflavone in Radix Astragali (RA), was found to upregulate Bcl-2 and downregulate Bax through activating the SIRT1/FOXO1/PGC-1α signaling pathway in OGD/R-induced hippocampal cells [188]. Similarly, Pic treatment significantly reduced the number of apoptotic hippocampal neurons in the cerebral ischemia/reperfusion injury (CIRI) mice model via activation of the Sirt1/FOXO1 pathway [172]. Kou et al. observed that magnolol could also increase the level of bcl-2 by activating SIRT1, thereby reducing brain edema and infarct volume and improving neurological scores [189]. According to Zhou et al.’s study, KFL reversed OGD-induced downregulation of SIRT1, concomitant with decreased caspase3, caspase9, and Bax, and enhanced Bcl-2 level [176]. Moreover, salvianolic acid B has also been proven to play a neuroprotective role by upregulating SIRT1 and Bcl-2 [190]. Stilbene glycoside is an ingredient extracted from *Polygonum multiflorum*. It upregulates SIRT3/AMPK expression, promotes mitochondrial autophagy, and inhibits cell apoptosis in ischemic neurons [191].

#### 3.1.6. Protecting the Blood–Brain Barrier

Donepezil, an acetylcholinesterase inhibitor, has been approved for the treatment of Alzheimer’s disease. In recent years, the neuroprotective effects of donepezil have attracted significant attention. By activating SIRT1 and further deacetylating FOXO3a and NF-κB, donepezil increases the expressions of tight junction proteins, indicating that donepezil may act as a key therapeutic agent of IS [192].

After a stroke, the serine/threonine-specific protein kinase CaMKK, a key kinase caused by increased intracellular calcium, maintains the integrity of BBB [193,194]. CaMKK phosphorylates the well-known vascular defender SIRT1, making it more stable and active [195]. In MCAO mice models, CaMKKβ activity inhibition reduced the pSIRT1 levels, reducing brain endothelial cell damage and ischemia-induced BBB leakage [196].

TLB could upregulate VEGFA protein and its receptor VEGFR-2, thereby promoting the proliferation of cerebral microvascular endothelial cells and angiogenesis in mice with CIRI [197]. The underlying mechanism may be that TLB binds to SIRT7, thus activating the SIRT7/VEGFA signaling pathway. Zhu et al. discovered that the vascular length, vascular density, branching index, and the structure of brain microvascular were significantly improved after notoginsenoside R1 (R1) treatment in MCAO rats [198]. It is possible that R1 exerts its effect by regulating the Nampt-NAD^+^-SIRT1 cascade and Notch/VEGFR-2 signaling pathway. In OGD/R-induced HBMEC cells, R1 activated the Nampt-NAD^+^-SIRT1 cascade and upregulated VEGFR-2 [198]. The activation of Notch signaling and VEGFR-2 is necessary for angiogenesis [199].

#### 3.1.7. Enhancing Neurogenesis

The Wnt signaling pathway is important for promoting neurogenesis and the development of neural tissues after cerebral ischemia [200]. Nuclear transfer and accumulation of β-catenin is an essential step in activating the Wnt/β-catenin signaling pathway [201]. Evidence suggests that SIRT1 promotes the deacetylation of β-catenin, resulting in β-catenin nuclear accumulation, which is essential for SIRT1 transcriptional activity [202,203]. *Momordica charantia*, an important multipurpose edible, medicinal plant, is the major source of bioactive compounds Momordica charantia polysaccharides (MCP). According to reports, MCP therapy promotes the differentiation of hippocampal SGZ neurons in MCAO rats, possibly by upregulating the SIRT1 level, thereby promoting the β-catenin deacetylation and nuclear accumulation [204,205]. This effect has also been confirmed through in vitro experiments.

In short, if SIRTs modulators with good pharmacokinetics and tolerance can be developed and converted into clinical applications, they can provide important treatment methods for IS. Therefore, research and development of effective drugs of SIRTs warrant urgent attention for IS treatment.

### 3.2. Non-Pharmacological Therapeutic/Rehabilitative Interventions

Acupuncture is an ancient traditional Chinese medicine technology, which is applied to treat various diseases, because of its advantages, including safety, potent therapeutic properties, low risks, and cost-effectiveness [206,207]. Electroacupuncture (EA) is an important form of acupuncture, in which it is easy to control and quantify the intensity and frequency of acupoint stimulation. It combines the advantages of acupuncture treatment and electrical stimulation and is widely used in clinical practice [208,209]. EA preconditioning significantly alleviates the neural damage caused by cerebral ischemia [210]. Ying’s study revealed that EA treatment at 24 h after ischemic stroke significantly inhibited the expression of autophagy-related proteins LC3II/I, Beclin1, and cell apoptosis, reducing brain injury in rats [211]. This neuroprotection may be achieved by promoting the expression of SIRT1, p-ERK, and p-JNK. However, autophagy is a double-edged sword in cerebral ischemia. The acetylation of H4K16 is associated with I/R-induced autophagic activation [212,213,214]. Another study showed that EA can trigger the molecular histone switch of SIRT1 and reduce the acetylation of H4K16, thus promoting the expression of LC3II/I and Beclin1 and reducing I/R damage [215].

In clinical trials, transcranial direct-current stimulation (tDCS), a non-invasive brain stimulation, is safe and with a neuroprotective benefit for stroke patients [216,217]. In the MCAO/R rat model, tDCs treatment has been found to upregulate SIRT6, thereby reducing DNA double strand-break and exerting neuroprotective effects [218].

The supplementary therapy known as hyperbaric oxygen therapy (HBO) is frequently used to treat various pathological disorders, mostly those caused by hypoxia and/or ischemic conditions. HBO can induce the deacetylation of HMGB1 by upregulating the level of SIRT1, thereby alleviating the neuroinflammatory reaction induced by OGD-R and improving ischemic brain injury [219]. In addition, HBO treatment also can increase the levels of ATP and NAD^+^ and consequently increase SIRT1 expression, leading to the attenuation of brain infarction volume, BBB integrity, and improvement of neurological functions in MCAO rats [220] (Table 1).

**Table 1 biomolecules-13-01210-t001:** Pharmacotherapy and non-pharmacological therapeutic/rehabilitative interventions targeting SIRTs.

Models/Subjects	Inducers	Therapeutics	Effects	Targets or Pathways	Reference
NAD^+^ enhancers					
Mice	CCAO	NMN	↓Mitochondrial fragmentation; ↓ROS	↑SIRT3	[168]
SIRTs inhibitors					
Mice	MCAO/R	AK-7	↑Neurological function	↓SIRT2; ↑p38	[185]
Cortical neurons	OGD	AK-1; AGK2	↓Apoptosis	↓SIRT2; ↓caspase3, Bim	[140]
SIRTs activators					
Mice	MCAO/R	Melatonin	↓Mitochondrial dysfunction	↑SIRT1	[165]
Hippocampal HT22 cells	OGD/R	Melatonin	↑Mitochondrial mass; ↑mitochondrial fusion/division kinetics	↑SIRT3	[166]
Rats	MCAO/R	Resveratrol	↓ATP energy consumption	↑cAMP/AMPK/SIRT1 pathway	[99]
SH-SY5Y cells	OGD/R	Notoginseng leaf triterpenes	↓Mitochondria dysfunction	↑Nampt-NAD^+^-SIRT1 pathway	[170]
Mice	MCAO/R	Picetannol	↓Oxidative stress	↑SIRT1/FOXO1; ↑SOD, CAT, GSH-Px	[172]
Rats	MCAO/R	Diosmetin	↓Oxidative stress	↑SIRT1/Nrf2; ↓LDH, ROS	[173]
Neuroblastoma cells	HR	Mangiferin	↓Oxidative stress	↑SIRT1/PGC-1α; ↓LDH, ROS	[174]
Mice	pMCAO	Alpha-lipoic acid	↓Oxidative stress	↑SIRT1/PGC-1α; ↑SOD	[175]
PC12 Cells	OGD/R	Kaempferol	↓Oxidative stress	↑SIRT1; ↓LDH, ROS; ↑SOD, GSH	[176]
Rats	MCAO/R	Betulinic acid	↓Autophagy	↑SIRT1/FOXO1; ↓Beclin 1, LC3I/II; ↑p62	[123]
Rats	MCAO/R	Magnoflorine	↓Autophagy	↑SIRT1/AMPK; ↓LC3; ↑p62	[177]
Mouse primary cortical microglia	OGD	MDL-811	↓Neuroinflammation	↑SIRT6/EZH2/FOXC1 pathway	[178]
Mice	pMCAO	Activator 3	↓Neuroinflammation	↑SIRT1; ↓NF-κb	[34]
Rats	MCAO/R	Ginkgo biloba	↓Neuroinflammation	↑SIRT1; ↓NF-κb	[180]
Rats	MCAO/R	Arctigenin	↓Neuroinflammation	↑SIRT1; ↓NLRP3, IL-1β, IL-18	[181]
Mice	MCAO/R	Bergenin	↓Neuroinflammation	↑SIRT1/FOXO3a; ↓NF-κb, IL-1β, IL-6, TNF-α	[182]
Mice	MCAO/R	Cycloastragenol	↓Neuroinflammation	↑SIRT1; ↓TNF-α, IL-1β	[183]
Rats	MCAO/R	Trilobatin	↓Neuroinflammation	↑SIRT3; ↓TLR4, NF-κb	[184]
Rats	MCAO/R	Resuvastatin	↓Apoptosis	↑SIRT1; ↓NF-κb	[142]
Mice	MCAO/R	Melatonin	↓Apoptosis	↑SIRT3	[186]
Rats	MCAO/R	Resveratrol	↓Apoptosis	↑SIRT1; ↓p53, caspase3	[187]
Hippocampal cell	OGD/R	Calycosin-7-O-β-D-glucoside	↓Apoptosis	↑SIRT1/FOXO1/PGC-1α; ↑Bcl-2; ↓Bax	[188]
Mice	MCAO/R	Picetannol	↓Apoptosis	↑SIRT1/FOXO1; ↓apoptotic hippocampal neurons number	[172]
Rats	MCAO/R	Magnolol	↓Apoptosis	↑SIRT1; ↑Bcl-2	[189]
PC12 Cells	OGD/R	Kaempferol	↓Apoptosis	↑SIRT1; ↑Bcl-2; ↓caspase3, caspase9, Bax	[176]
Rats	MCAO/R	Salvianolic acid B	↓Apoptosis	↑SIRT1; ↑Bcl-2	[190]
Rat PC12 cells	OGD/R	Stilbene glycoside	↓Apoptosis; ↑mitochondrial autophagy	↑SIRT3/AMPK pathway	[191]
HBMECs	OGD/R	Donepezil	↑BBB integrity	↑SIRT1; ↓FOXO3a, NF-κb; ↑TJ proteins	[192]
Mice	MCAO/R	CaMKK	↑BBB integrity	↑SIRT1	[196]
Mice	MCAO/R	Trilobatin	↑Angiogenesis	↑SIRT7/VEGFA pathway	[197]
Rats	MCAO/R	Notoginsenoside R1	↑Angiogenesis	↑NAMPT-NAD^+^-SIRT1 pathway; ↑Notch/VEGFR-2 pathway	[198]
HBMECs	OGD/R	Notoginsenoside R1	↑Angiogenesis	↑NAMPT-NAD^+^-SIRT1 pathway; ↑Notch/VEGFR-2 pathway	[198]
Rats	MCAO/R	Momordica charantia polysaccharides	↑Neurogenesis	↑SIRT1; ↑β-catenin nuclear accumulation	[204,205]
C17.2 cells	Glutamate	Momordica charantia polysaccharides	↑Neurogenesis	↑SIRT1; ↑β-catenin nuclear accumulation	[204]
C17.2 cells	OGD	Momordica charantia polysaccharides	↑Neurogenesis	↑SIRT1; ↑β-catenin nuclear accumulation	[205]
**Non-pharmacological therapeutic/rehabilitative interventions**					
Rats	MCAO/R	EA	↓Autophagy, apoptosis	↑SIRT1; p-ERK, p-JNK; ↓ Beclin 1, LC3I/II; ↑p62; ↓apoptotic cells	[211]
Rats	MCAO/R	EA	↑Autophagy	↑SIRT1; ↑Beclin 1, LC3I/II;	[215]
Rats	MCAO/R	tDCS	Neuroprotective effect	↑SIRT6; ↓DNA double strand-break	[218]
N2a cells	OGD/R	HBO	↓Neuroinflammation	↑SIRT1; ↓HMGB1	[219]
Rats	MCAO/R	HBO	↑BBB integrity, neurological functions; ↓brain infarction volume	↑NAD; ↑SIRT1; ↑ATP	[220]

Abbreviations: middle cerebral artery occlusion (MCAO); middle cerebral artery occlusion/reperfusion (MCAO/R); common carotid artery occlusion (CCAO); oxygen–glucose deprivation/reperfusion (OGD/R); hypoxia 12 h/reoxygenation 12 h (HR); nicotinamide mononucleotide (NMN); permanent middle cerebral artery occlusion (pMCAO); rat pheochromocytoma (PC12); human neuroblastoma cells (SH-SY5Y cells); brain microvascular endothelial cells (HBMECs); electroacupuncture (EA); transcranial direct-current stimulation (tDCS); hyperbaric oxygen therapy (HBO); inhibition (↓); activation (↑).

### 3.3. Epigenetic Regulators

Epigenetic modification is a heritable change in gene expression and regulation that does not involve DNA sequence changes. These modifications can result in the loss of protein function, changes in body structure and function, and disease occurrence. The modification mechanism of epigenetics mainly includes DNA methylation, histone modification, and regulation of non-coding RNAs (ncRNAs). NcRNAs, which primarily comprise microRNAs (miRNA), long non-coding RNAs (lncRNA), and circular RNAs (circRNA), can influence gene expression at both the post-transcriptional and translational stages [221]. In this section, we mainly focus on the regulation of SIRTs by ncRNAs.

MiRNAs are endogenous non-coding RNAs about 20–25 nucleotides in length, which usually interact with their downstream target, a short sequence of mRNA’s 3′-UTR region, to either inhibit its translation or cause its degradation to regulate its expression [222]. According to Ruan et al.’s study, miR-370 was upregulated in brain tissues of MCAO rats and its knockdown decreased the volume of cerebral infarction, blocked cell apoptosis, and promoted the expression of Nrf2 and HO-1 in vivo [223]. Further investigation revealed that SIRT6 is miR-370’s direct target, and overexpressing SIRT6 partially counteracted the effect of miR-370 on OGD/R-induced SH-SY5Y cell death by activating the Nrf2 signaling pathway [223]. In addition, miR-19a/b-3p expression is significantly upregulated during cerebral ischemia/reperfusion injury [224]. Further experimental evidence demonstrated that the upregulated miR-19a/b-3p promotes the occurrence and development of inflammation during cerebral ischemia/reperfusion injury. MiR-19a/b-3p inhibits the expression of SIRT1, thereby activating FOXO3, which in turn activates the SPHK1 signaling pathway. Activation of this pathway subsequently increases the production of inflammatory factors and infiltration of inflammatory cells, exacerbating cerebral ischemia/reperfusion injury [224].

LncRNAs are non-coding RNAs with a length of more than 200 nucleotides but with no protein-coding function [225]. Experiments have shown that lncRNAs participate in the regulation of the posttranscriptional process through miRNA sponging [226]. LncRNAs are mainly involved in the pathogenesis of ischemic stroke through pathological processes, including oxidative stress, apoptosis, BBB permeability, and inflammatory response. LncRNA nuclear enriched abundant transcript 1 (NEAT1) upregulates SIRT3 by targeting the expression of mitofusin 2 (Mfn2), thereby exerting anti-oxidative stress and apoptosis effects caused by ischemia-reperfusion [227]. Small nucleolar RNA host gene 15 (Snhg15), a lncRNA, is a sponge of miR-141. Snhg15 was significantly upregulated in OGD/R-induced SH-SY5Y cells, and based on this finding, Kang discovered that Snhg15 may indirectly upregulate SIRT1 by inhibiting miR-141, resulting in a decrease of ROS, iNOS, IL-1β, and IL-6 [228]. In addition, the expression of Snhg7 was upregulated in the OGD/R-induced PC12 cells [229]. And Snhg7 overexpression could attenuate oxidative stress and cell damage brought on by OGD/R. The results of further research showed that Snhg7 may interact with miR-9 to restrict its expression while increasing the expression of SIRT1, which in turn reduced ROS generation, MDA level, and apoptotic rate [229]. Tian and colleagues revealed that Snhg8 could inhibit the inflammatory response of microglia and BBB injury by regulating the SIRT1-mediated NF-κB pathway by sponging miR-425-5p [230]. Additionally, Snhg8 inhibits microglia activation and BBB permeability by sponging miR-449c-5p to upregulate SIRT1 and FOXO1, with neuroprotective effects against ischemic brain injury [231]. Moreover, Xu et al. revealed that lncRNA H19 is markedly downregulated in MCAO mice and OGD-induced HT22 cells [232]. Knockdown of lncRNA H19 reversed the decline in OGD-induced miR-29b, SIRT1, and PGC-1α levels, causing a decrease in apoptotic cells and inflammatory cytokine concentration [232].

CircRNAs are a new class of ncRNAs characterized by a closed-loop structure. They help in the expression of functional genes by binding with corresponding microRNAs or directly interacting with proteins [233]. Chen et al. reported that circHIPK3 (circRNA homeodomain-interacting protein kinase 3) functions as an endogenous sponge of miR-148b-3p, leading to the downregulation of SIRT1. This study observed the overexpression of circHIPK3 in the brain tissue of MCAO mice and identified apoptosis and mitochondrial dysfunction in BMECs [234]. Their findings indicate that circHIPK3 may play a role in the pathogenesis of cerebral ischemia by modulating the miR-148b-3p/SIRT1 axis. Moreover, experiments by Yang et al. demonstrated that the expression of circ-Rps5 alleviated LPS-induced inflammation and apoptosis through the NF-κB signaling pathway by increasing the SIRT7 level. A further luciferase reporter revealed that the circ-Rps5 directly targeted the miR-124-3p. Therefore, it is reasonable to speculate that circ-Rps5 attenuated ischemic-stroke-induced brain injury via the miR-124-3p/SIRT7 signaling pathway [235] (Figure 3).

### 3.4. Potential Molecules Targeting SIRTs

Besides the above treatment strategies, other molecules that can target SIRTs to confer neuroprotective effects in IS have been reported.

In OGD/R-induced hippocampal neuron cells, overexpression of C1q/tumor necrosis factor-related protein-3 (CTRP3) suppressed cell apoptosis and promoted mitochondrial biogenesis and physiological functions through a mechanism involving activation of the AMPK/SIRT1/PGC-1α pathway [236].

TGR5 is a plasma-membrane-bound G protein-coupled bile acid receptor [237]. Liang et al. reported that activation of TRG5 increased the tight junction (TJ) protein expression thereby improving the disrupted BBB in the ischemic brain via the BRCA1/Sirt1-related signaling pathway [238]. BRCA1 is a tumor suppressor gene expressed by endothelial cells and is a key regulator of SIRT1 [239,240,241].

Interferon (IFN) regulatory factors (IRFs) are transcription factors known to exert immunomodulatory roles by altering the expression of interferon genes. IRF9 was reported to be a negative transcriptional regulator of SIRT1. Chen and colleagues discovered that IRF9 is specifically activated in neurons of mice brains and this exacerbates brain injury upon I/R insult. The principal pathological effect of IRF9 is the downregulation of SIRT1 transcriptional expression, which leads to the activation of p53. This, in turn, results in an elevated number of TUNEL-positive cells and an increase in caspase3 levels [242].

The proto-oncogene c-myc encodes the transcription factor c-Myc, which is important for regulating cell growth and viability. A crucial regulator known as Myc coordinates various cell signals and mediates transcriptional processes. Upregulation of c-Myc can partially resolve the motor impairment induced by cerebral ischemic stroke [243]. Liu et al. discovered that c-Myc protects MCAO mice from neuronal damage by increasing miR-200b-5p-regulated SIRT1 expression [244].

Furthermore, overexpression of lanthionine synthetase C-like protein 1 (LanCL1) significantly reduced the release of LDH, increased the mitochondrial enzyme activity, and attenuated apoptosis in OGD-induced neuronal HT22 cells in a SIRT3-dependent manner [245]. This indicates that LanCL1 is a promising target for the therapy of IS.

### 3.5. Stem Cell-Derived Exosome Therapy

Research has found that the exosomes generated by bone marrow mesenchymal stem cells are rich in the transcription factor early growth response protein 2 (Egr2), which has the potential to treat IS [246]. Egr2 inhibits the activation of the Notch signaling pathway by upregulating the expression of SIRT6, thereby reducing MCAO/R-caused brain damage, and promoting angiogenesis in OGD/R-treated brain endothelial cells. This study provides a potential exosome-based therapy targeting SIRT6 for IS [246]. However, further research and clinical trials still need to verify the safety and effectiveness of this treatment strategy.

## 4. Concluding Remarks and Prospects

Limited therapeutic strategies are currently available to address ischemic stroke, hence it is a significant public health and socio-economic challenge. SIRTs are potential candidates for identifying novel molecular pathways and target molecules for drug development of ischemic stroke. Although David T. She reviewed the role and therapeutic significance of sirtuins in IS in 2017, this report updates research progress in recent years [247]. The broad introduction to SIRTs and IS in this review includes information on their expression, subcellular location, enzymatic action, and regulatory involvement in IS. The high expression of SIRTs in the brain and multiple targets changes multiple biological processes in response to ischemic stimuli, including regulating energy metabolism, inhibiting oxidative stress, modulating autophagy, mitigating inflammatory response, anti-apoptosis, protecting BBB integrity, promoting angiogenesis and enhancing neurogenesis. Further, we described the roles of SIRTs in the mechanisms of various treatment modalities including medical treatment, non-pharmacological therapeutic/rehabilitative interventions, epigenetic regulators, molecules regulation, and stem-cell-derived exosome therapy (Figure 4). Altogether, our results indicate that the sirtuin family holds great therapeutic potential in the treatment of IS.

In the development of strategies targeting sirtuins in ischemic stroke, it is important to consider the potential drawbacks. One major limitation is that the majority of SIRTs studies only focus on neurons or endothelial cells. However, the neurovascular unit (NVU) forms the fundamental building block of brain tissue, and the pathophysiology and neurovascular repair process of ischemic stroke are closely linked to the NVU’s steady-state [248]. Therefore, a treatment scheme that only targets neurons or endothelial cells alone and ignores the interaction of various nerve cells after cerebral ischemia injury may not be therapeutically effective. Secondly, SIRTs regulate several systemic diseases, and different SIRTs may exert different effects on various diseases, or even opposite outcomes [10]. Therefore, there is a need to investigate strategies to develop SIRT as a personalized therapeutic target for IS. In summary, although we have provided an overview of some pharmacological SIRT modulators, current research has not sufficiently addressed their pharmacokinetics, efficacy, and safety. Future research should fill these gaps and actively investigate the development of modulators that can cross the blood–brain barrier. To achieve this, a number of important questions need to be answered. For example, (1) SIRT1 has been widely studied, but the exact function and therapeutic effect of other SIRTs in IS are still unclear, and (2) is the role of SIRTs consistent between acute and chronic cerebral ischemia? In conclusion, despite the considerable amount of preclinical evidence presented on the role of sirtuins in ischemic stroke, the understanding of their clinical significance is limited. Therefore, additional clinical trials are necessary to more comprehensively investigate the potential of sirtuins as a viable treatment for ischemic stroke.

## Figures and Tables

**Figure 1 biomolecules-13-01210-f001:**
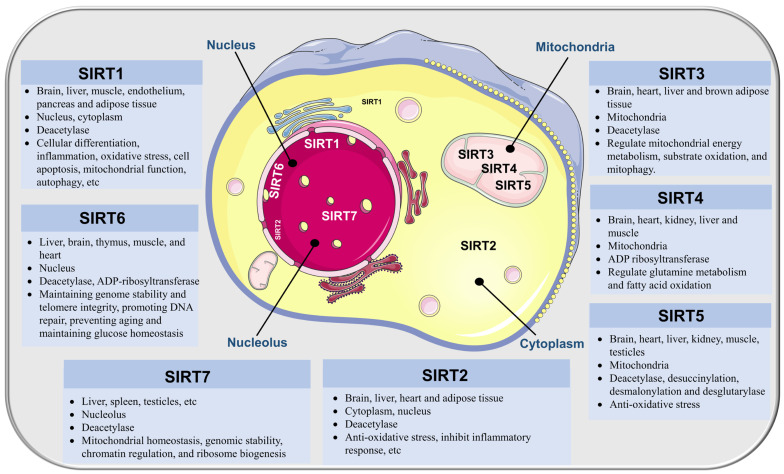
Distribution, subcellular location, enzyme activity, and functions of SIRTs.

**Figure 2 biomolecules-13-01210-f002:**
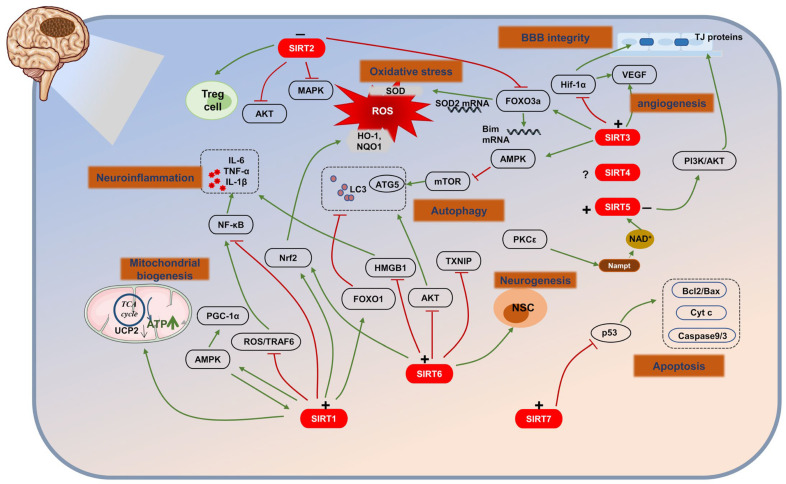
The role of SIRTs in ischemic stroke (IS). (+) means activation; (−) means inhibition. SIRTs, by acting on multiple targets, regulate several pathological processes including energy metabolism, oxidative stress, autophagy, neuroinflammation, apoptosis, BBB disruption, angiogenesis, and neurogenesis.

**Figure 3 biomolecules-13-01210-f003:**
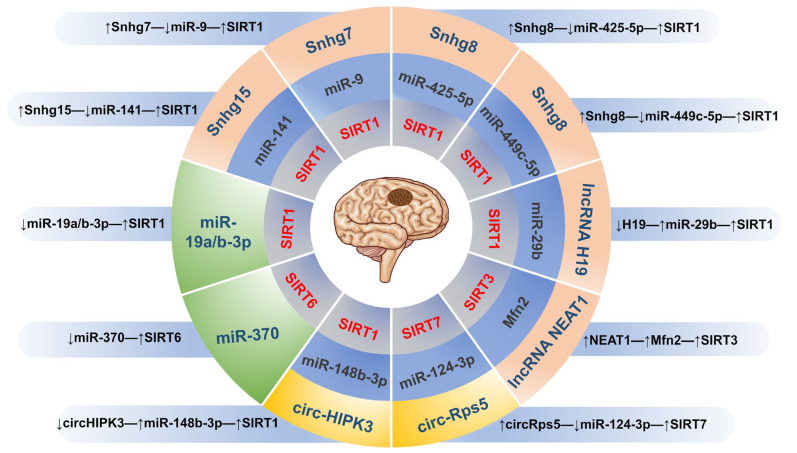
The epigenetic regulatory mechanism of targeting SIRTs in IS. (↓) means inhibition; (↑) means activation.

**Figure 4 biomolecules-13-01210-f004:**
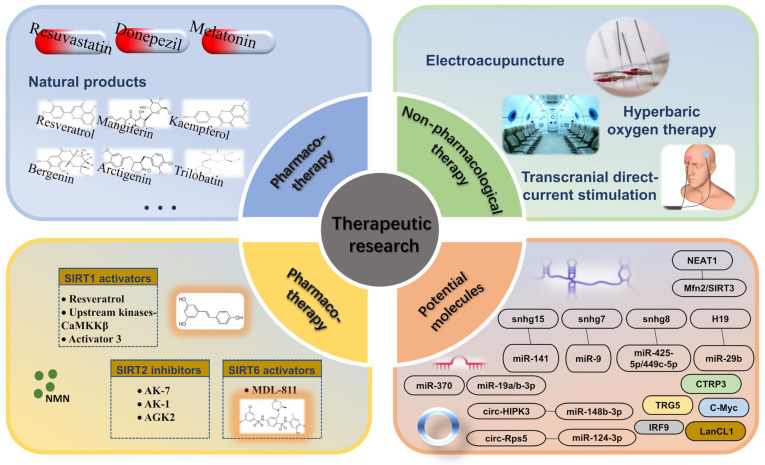
Overview of IS treatment strategies targeting SIRTs.

## Data Availability

Not applicable.

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
