# Peer review of "Sirtuins: Promising Therapeutic Targets to Treat Ischemic Stroke"

_biomolecules, 2023, doi:10.3390/biom13081210_

Round 1
Reviewer 1 Report
In this review, Liu et al discussed the function of sirtuins in the pathophysiological processes of ischemic stroke and the therapeutic implications. Although a thorough review authored by David T. She has summarized this topic in 2017, the current one contains some latest advances. This paper will be of interest to a broad spectrum of neuroscientists who are interested in the functional role of sirtuins in stroke. Several suggestions are listed below.
1. In the part 3 related to the therapeutics, the author mainly focused on the SIRT1. The therapeutic implications of other sirtuins should be added.
2. It is necessary to mention the previous review (She et al., 2017) and indicate the difference between them.
3. Although sirtuins are expressed in human brain in the introduction, no one clinical study related to the role of sirtuins in stroke was mentioned in the following parts.
4. The author mainly talked about the function of sirtuins in MCAO models in vivo or OGD model in vitro. Expect for those, other ischemic models are needed to be included.
Minor editing of English language required
Author Response
Dear reviewer,
We would like to thank you for your time and effort spent in reviewing our manuscript. The careful reading, helpful comments, and constructive suggestions, which has significantly improved the presentation of our manuscript. We have tried our best to revise the manuscript according to your kind and constructive comments and suggestions. Revised portions are marked in yellow in the paper. Additionally, we have improved the language expression of the paper to facilitate better reading and understanding for the readers. In this document, we will respond to your comments point by point.

Reviewer 2 Report
This review provides extensive insight into the role of sirtuins, a class of NAD+-dependent deacylases, in the pathophysiology of stroke and their potential as therapeutic targets. Despite promising results in experimental stroke studies showing beneficial effects upon modulating sirtuin-related signaling pathways, this is the first review gathering evidence associating the activities of sirtuins and stroke, other than reviews focusing on a particular sirtuin in stroke. Although there has been an increasing interest in this field in the past years, its therapeutic potential might have been overlooked. The urgent need to develop novel and effective treatments for a devastating disease as stroke places this work as a relevant review article in the field. However, there are major aspects that should be addressed to provide a thorough, comprehensive and accurate understanding, before being considered for publishing.
1. Key concepts/terminology
Some paragraphs are worth reformulating for a better understanding, such as the fragment between lines 121-129. L604-607 Some of the phrases are long and/or hard to follow, such as in lines 604-607 (““Chen et al. reported…, … in brain microvascular endothelial cells (BMECs)”.
Some of the terminology used is worth reviewing such as ‘thrombectomy’, more common for ‘embolectomy’ (line 28).
In other sections, the information exposed can lead to erroneous statements; the text should be rigorously reviewed accordingly. For example, in lines 34-35 histone modifications being described as a cause of ischemic stroke, or in line 158 describing a study reporting a reduction in infarct volume in wild-type mice exposed to MCAO (is this versus the SIRT5 KO animals? If so, it would be the KO condition altering the infarct size seen in WT conditions). Also, Bax is described as anti-apoptic in line 336.
The association SIRT1-TRAF6 seems vague, not well supported by literature but connecting indirect findings in the way it is described in the text (lines 300-314). Could you please justify the relevance for this association?
Some misconception has also been detected with key pathophysiological mechanisms associated with stroke that warrant revisiting. Angiogenesis is described to reduce the leakage of the BBB after stroke (lines 380-381), yet BBB leakage is a downside/necessary process for angiogenesis (i.e. in destabilizing endothelial cells junction for tip cells to migrate in vessel sprouting). Please review these concepts throughout the text.
The term ‘Nerve regeneration’ (line 399) does not accurately reflect the content described in this section, please reconsider the terminology.
Using the term ‘Physical therapy’ (title in line 535) leads to confusion with ‘physiotherapy’, which is a different concept to the therapeutic approaches mentioned in this paragraph. Appropriate terminology or restructuring should be considered.
There are some key concepts that need revisiting, such as using the term CIRI as a stroke model. In Table 1, it is used as a distinct model to MCAO in the category ‘Inducers’. However, the model used in the cited studies is MCAO, whereas CIRI referring to the pathophysiological injury caused by ischemia/reperfusion within the model. In line with this, what is the difference between the terminology pMCAO and MCAO in Table 1, if there is also a MCAO/R term (presumably transient ischemia followed by reperfusion)? Could you please also explain other acronyms that are not detailed in the text nor Table legend, such as ‘HR’.
2. Structure
Following the same classification/structure of biological processes in the second section focusing on the role of SIRTs in IS as in the third section focused on therapeutic research, presents the information in a helpful, clear and concise way for the reader to associate the concepts presented herein. However, there are other structural aspects that should be carefully addressed.
In the section ‘Pharmacotherapy’ (line 425), dedicating a paragraph exclusively to Western medicine could be confusing as this would differentiate it from a holistic approach in Eastern medicine, which is not stressed nor mentioned thereafter and does not seem to meet the scope of this review. The following ‘Natural products’ section is scarce and the following sections seem to be separate. As a suggestion, could these ‘Pharmacotherapy’ subsections be retitled together and perhaps mention that this review focuses primarily on natural products targeting SIRTs-mediated pathways. As it is now, the sections divided by the processes targeted by natural products seem like a separate section to “Natural products” and is hard to follow the structure.
It is also confusing to have a separate “SIRTs modulators” section, the information of which could be included in the previous Therapeutic research “Pharmacotherapy” section. As a suggestion, could the sections be restructured and instead specify in the summary table whether these are natural compounds, direct SIRT modulators or compounds that indirectly modify sirtuins.
3. Additional aspects to cover
Despite the extensive review covering all sirtuin subtypes, location and function in relation to stroke, this review could benefit and improve by adding or developing further some of the following aspects.
- The predominance of different SIRTs in the brain or cell types within the brain, as well as the association between their biodistribution/cell type/subcellular location and their specific functions (i.e. the subcellular location of SIRT1 in apoptosis, with its nuclear but not cytoplasmic overexpression being related to anti-apoptosis)
- The role/importance of SIRTs specifically in acute or long-term stages post-stroke.
- Despite SIRT3-4-5 being considered as the ‘mitochondrial’ sirtuins, these are omitted in the sections covering Energy metabolism in ‘Regulation of Sirtuins in ischemic stroke’ and Mitigating mitochondrial injury in ‘Pharmacotherapy’.
- A bit more detailed description of the enzymatic activity/mechanism of SIRTs, rather than an extensive explanation of more basic concepts such as ‘epigenetics’.
4. References
The literature search for this review is remarkably extensive, however it should be highlighted that more than half of the cited references are more than 5 years old, considering that most of SIRT-stroke papers have been published in the past 5 years. Perhaps this could be updated.
5. Figures
The figures nicely summarize in a clear and visual way the content of this review. However, the resolution could be improved. Please find some suggestions below:
Fig 1
The consistency text could be improved to clarify concepts and standardize the structure, such as the terminology used for different tissues.
SIRT1/2 which can be present in different subcellular locations (cytoplasm, nucleus): instead of showing just the predominant location, why not place them in both but maybe with larger font in the predominant compartment?
Title: ‘location’ for ‘subcellular location’ to avoid confusion with distribution.
Fig 2
The title of the processes could be more visual (maybe changing color), and the arrows indicating processes being activated/inhibited could also be depicted in different colors to make it more visual.
Activation(+) or inhibition(-) of SIRTs: is this referred to as after IS? If so, please specify
The molecules that are influenced by SIRTs located in cytoplasm lead to confusion. It could help perhaps not focusing on the subcellular compartments as it is only depicted to show the predominant location of SIRTs, which was already done in Fig 1.
Fig 4
The orientation of the titles below makes it difficult to read.
The English language is understandable, however it warrants moderate reviewing to improve the overall quality, particularly in the grammatical form of expressions.
Author Response
Dear reviewer,
We sincerely thank you for your valuable feedback that we have used to improve the quality of our manuscript. As you are concerned, there are several problems that need to be addressed. We tried our best to improve the manuscript and made some changes to the manuscript. Revised portions are marked in blue in the paper. In addition, this manuscript has been edited for proper English language, grammar, punctuation, and spelling by one or more highly qualified native English-speaking editors. In this document, we will respond to your comments point by point.

Round 2
Reviewer 2 Report
I appreciate the authors’ response and revision. The new version of the manuscript has been improved and the authors have addressed previous concerns. I still have some observations from this second version that I believe should be addressed prior to publication:
Abstract: the term 'physical therapy' was updated in the main text but has not been corrected accordingly in the abstract and could lead to confusion.
Line 29: 'mechanical embolectomy' was not replaced with 'mechanical embolectomy', perhaps due to misexplaining on my behalf.
Line 156: I appreciate the effort in reformulating sentences such as this. However, different studies (references 71, 72) are referenced where explaining only one of these. And more importantly, there was a misconception regarding reference 71. This study shows no differences in infarct size when comparing SIRT5 KO vs. WT mice after 85 min MCAO. They do show that the neuroprotective effect of ΨεRACK (PKCε activator), as seen in a reduction of infarct size in the WT group, is not seen in the SIRT5 KO mice, therefore suggesting that the PKCε-mediated neuroprotection depends on SIRT5. In this light, the conclusion that SIRT5’s regulatory effect may be related to the duration of middle cerebral artery occlusion is not supported by these studies.
Fig 2: there is an incomplete process/spare arrow from SIRT1.
Fig 4: the titles are now clear and easy to read, however the quality of this figure can be further improved - most of the text is outside the boxes in 'potential molecules'
I appreciate the work done by the authors in improving the quality of English and clarity of the text in this new version. There are some minor errors in the text such as in line 30 that require proofreading.
Author Response
Dear reviewer,
We sincerely thank you once again for patiently reviewing our manuscript and providing constructive feedback. We have made modifications and adjustments to the manuscript based on your suggestions. The revised portions are highlighted in blue. In the Word document, we will respond to your comments point by point.
